# Boron Fertilization Alleviates the Adverse Effects of Late Sowing in Wheat under Different Tillage Systems

Muhammad Ijaz [1,2], Sami Ul-Allah [1,3], Ahmad Sher [1,2], Abdul Sattar [1,2], Khalid Mahmood [4,5], Saud Alamri [6,*], Yasir Ali [7], Farhan Rafiq [1], Syed Muhammad Shaharyar [1], Bader Ijaz [1] and Ijaz Hussain [1]

[1] College of Agriculture, University of Layyah, Layyah 31200, Pakistan; muhammad.ijaz@bzu.edu.pk (M.I.); samipbg@bzu.edu.pk (S.U.-A.); ahmad.sher@bzu.edu.pk (A.S.); abdulsattar04@gmail.com (A.S.); farhanbzu13@gmail.com (F.R.); syedsharyarly@gmail.com (S.M.S.); baderzaildar@gmail.com (B.I.); ijazhussain.uam3702@gmail.com (I.H.)

[2] Department of Agronomy, Bahauddin Zakariya University Multan, Multan 60000, Pakistan

[3] Department of Plant Breeding and Genetics, Bahauddin Zakariya University Multan, Multan 60000, Pakistan

[4] Nordic Seed/AS, Grindsnabevej, 25, 8300 Odder, Denmark; khma@nordicseed.com

[5] Department of Agroecology, Faculty of Science and Technology, Aarhus University, Forsogsvej 1, Flakkebjerg, 4200 Slagelse, Denmark

[6] Department of Botany and Microbiology, College of Science, King Saud University, Riyadh 11451, Saudi Arabia

[7] Department of Plant Pathology, College of Agriculture, University of Layyah, Layyah 31200, Pakistan; yasirklasra.uca@gmail.com

* Correspondence: saualamri@ksu.edu.sa

**Abstract:** Wheat (*Triticum aestivum* L.) is a staple and the most important food crop around the world. The growth and productivity of wheat are influenced by different factors, viz., sowing time, tillage system and nutrient application. The current field experiment consists of different boron (B) application rates, viz., $B_0$ = No application (Control), $B_1$ = soil applied (2 kg ha$^{-1}$), $B_2$ = foliar applied (2 kg ha$^{-1}$), $B_3$ = water spray; two tillage systems, viz., zero tillage (ZT) and conventional tillage (CT); and three sowing dates ($S_1$ = 15 November; $S_2$ = 5 December and $S_3$ = 25 December). It was conducted during the years 2019–2020 and 2020–2021 under a split-split plot arrangement. The results showed that sowing dates and boron had beneficial impacts on the growth and productivity of wheat. The wheat crop sown on 15 November showed the highest plant height, chlorophyll contents, grains per spike, and grains' boron content. Similarly, the application of boron under late sown conditions also improved the plant height (83.8 cm), chlorophyll contents (45.6), biological (5418 kg ha$^{-1}$) and grain (4018 kg ha$^{-1}$) yield as compared to control during both years. Furthermore, the higher crop growth and yield parameters were noted with the foliar application of boron at 2 kg ha$^{-1}$. However, wheat crop growth and yield characteristics were not significantly affected by tillage techniques, h. In conclusion, the application of boron @ 2 kg ha$^{-1}$ could be a suitable option for achieving higher wheat grain yield and productivity under late-sown conditions.

**Keywords:** sowing dates; boron fertilization; *Triticum aestivum*; foliar fertilization; zero tillage; late sowing; grains boron contents; grain yield

## 1. Introduction

Wheat (*Triticum aestivum* L.) is a fundamental staple food crop globally and major source of carbohydrates and energy for human consumption [1,2]. Wheat seeds are composed of 2.11%, 68%, 2.9% and 15.4% minerals, carbohydrates, fats and proteins, respectively [3]. However, the demand for wheat is increasing constantly due to the ever-increasing population, and it will rise almost 60% by the year 2050 [4]. To feed the growing global population and end famine, an increase in wheat crop yield is essential. However, several factors such as improper fertilizer management, improper planting times, delayed cotton harvesting, and a scarcity of high-yielding cultivars lead towards reduced wheat

crop growth and productivity [5,6]. In this context, sowing time is a very crucial factor for sustainable crop growth and productivity. The optimal sowing time results in increased productivity without increasing expenses [7,8]. Optimum sowing of the wheat crop results in increased tillering, number of spikes, grain weight, and ultimately an enhanced crop yield [9,10]. Similarly, the wheat crop being sown too early results in weak tillers with a depleted root system as a result of high temperatures, which also causes poor germination and embryo mortality. On the other hand, a low temperature will result in poor stand establishment, reduced tillering and ultimately declined crop yield in late-sown crops [11,12].

Furthermore, the late-sown wheat crop also has a heavy susceptibility to diseases, which results in poor crop yield [13]. Additionally, the late-sown wheat crop is more prone to the damaging impacts of heat stress at the crop grain filling stage, which decreases grain size, grain quality and grains' yield [14,15].

Boron (B) is one of the most important micronutrients that plants need for healthy growth and development [16]. However, boron deficiencies affect 31% of the world's arable land and 49% of the arable land in Pakistan [17]. Sowing wheat crops in boron-deficient soil leads to grains sterility in the wheat crop [18]. Similarly, the boron deficiency leads towards cell wall breakdown, slowed proton pump, ATPs development, lower photosynthetic productivity, and declined electron transport chain functioning [19,20]. Boron also affects several biological functions in plants, including the synthesis of cell walls, proteins, the metabolism of carbohydrates and nucleic acids and cell wall enlargements, and assimilates translocation [21]. The application of boron also improves the uptake of other mineral nutrients (N, P, K) by the plants [18]. Despite having minimal direct impact on photosynthetic activities, the availability of boron results in increased activities of net photosynthetic rates through capturing different plant pigments such as carotenes and chlorophyll in plants' photosynthetic structures [22,23]. Furthermore, the application of boron at the reproductive stage boosts wheat crop grain yield [24]. Additionally, boron also helps to tolerate the alarming effects of high temperature stress in the late-sown wheat crop [25]. In previous studies, it has been explained that boron application (1.6 kg ha$^{-1}$) improved the wheat grain yield and yield components [26]. It was also explained by [26] that the application of higher doses of boron (4 kg ha$^{-1}$) decreased the quantity of grain yield and also the yield components. Boron fertilization with 4 kg ha$^{-1}$ caused a trend towards a lower density of ears per unit area, a lower number of grains per air and lower thousand grain weight [26]. Several studies have also reported that the application of boron improves the wheat crop growth and productivity by improving the crop metabolism, nutrients uptake and protein synthesis [18,19,23]. Furthermore, the application of boron in crop plants improved the growth and productivity in wheat, lentil and chickpea [27–29]. Boron application also helps the wheat crop to tolerate heat stress conditions due to delayed sowing [30].

Boron is a versatile micronutrient that is used in the production of both perennial and seasonal crops. It is often applied in the form of compacted fertilizers in either the spring or fall seasons. The substance can be applied by either directly adding it to the soil or spraying it on the leaves. In some cases, seed dressing may also be used [18]. The presence of this element is critical for plants throughout their whole life cycle, beginning with germination and continuing until maturity. Boron is an essential element that plays critical roles in a variety of plant processes, including calcium utilization, cell division, water relations during generative growth, disease resistance and nitrogen metabolism. Boron, the only essential element for plants, is absorbed as an uncharged molecule rather than an ion [16].

Boron is a vital element that is required for plant growth and development. Boron's various metabolic, nutritional, hormonal and physiological roles have recently been explained, emphasizing its importance for both human and animal health [21]. Ongoing research is being conducted to determine the impact of this component on metabolic processes. According to research, a lack of calcium and fluorine may cause metabolic dysfunctions, which might present as arthritis, osteoporosis and cerebral anomalies. Boron

is a micronutrient that people and animals receive via their everyday diet. The amount of boron obtained is determined by the dietary composition [19].

Tillage is considered as natural mulch that helps to attain maximum crop growth and productivity. The majority of studies on tillage systems have focused on soil carbon (C), nitrogen (N), and phosphorous (P) changes. Therefore, wheat sowing with minimum tillage systems enhances the soil organic carbon [10]. The systemic absorption of micronutrients in the soil profile is affected by the tillage operations [31]. The use of minimum tillage practices is supposed to ensure timely planting, maintain the planting accuracy and cost savings [32,33], and improve and sustain the crop quality and yield [34]. Zero tillage (ZT) systems are more effective than the conventional tillage system (CT) in terms of carbon sequestration and reduced $N_2O$ and GHGs emissions [35]. The soil's physical and chemical properties are also improved under minimum tillage practices. However, it takes a longer time period (4–5 years) to improve soil health and productivity through the decomposition of organic matter and crop residues [10]. Furthermore, zero tillage practices result in improved crop water use efficiency by improving the water holding capacity of soil, and in reduced water losses through runoff and evaporation [36], while conventional tillage practices lead to the loss of soil organic matter and nutrients due to enhanced soil erosion [37,38].

Hence, keeping in mind the above-mentioned facts, it is clear that crop residues contain higher concentrations of applied nutrients, which are returned back to the soil after decomposition under conservation tillage practices [39]. Therefore, the adaptation of conservation tillage practices and optimum boron applications could be a sustainable practice in improving soil fertility and crop production under late-sown conditions. Moreover, different studies have been conducted to evaluate the effect of mineral nutrients (N, Zn, S) on crop growth and productivity when sown at optimal and late conditions under conservation tillage systems [10,35]. However, there has been limited research work reported on the effects of the optimized application of boron in wheat crops under late-sown conditions with different tillage systems. Therefore, a field study was conducted to examine the positive effects of varying doses of boron on the growth and productivity of wheat crops. The study was conducted under different tillage systems and included both optimal and late sowing conditions.

## 2. Materials and Methods

### 2.1. Location and Soil

This field trial was carried out at BZU, Bahadur Sub Campus, Layyah (Figure 1). The experimental location was situated in the Thal region of Punjab-Pakistan. Soil was sandy loam including 11% clay, 39% silt and 50% sand. Additionally, the soil's physio-chemical properties were determined prior to conducting the field experiment. The soil composition consisted of organic matter (0.74%), soil available nitrogen, phosphorous, potassium at 69.25, 7.35 and 205 mg kg$^{-1}$, respectively, and soil pH (8.00).

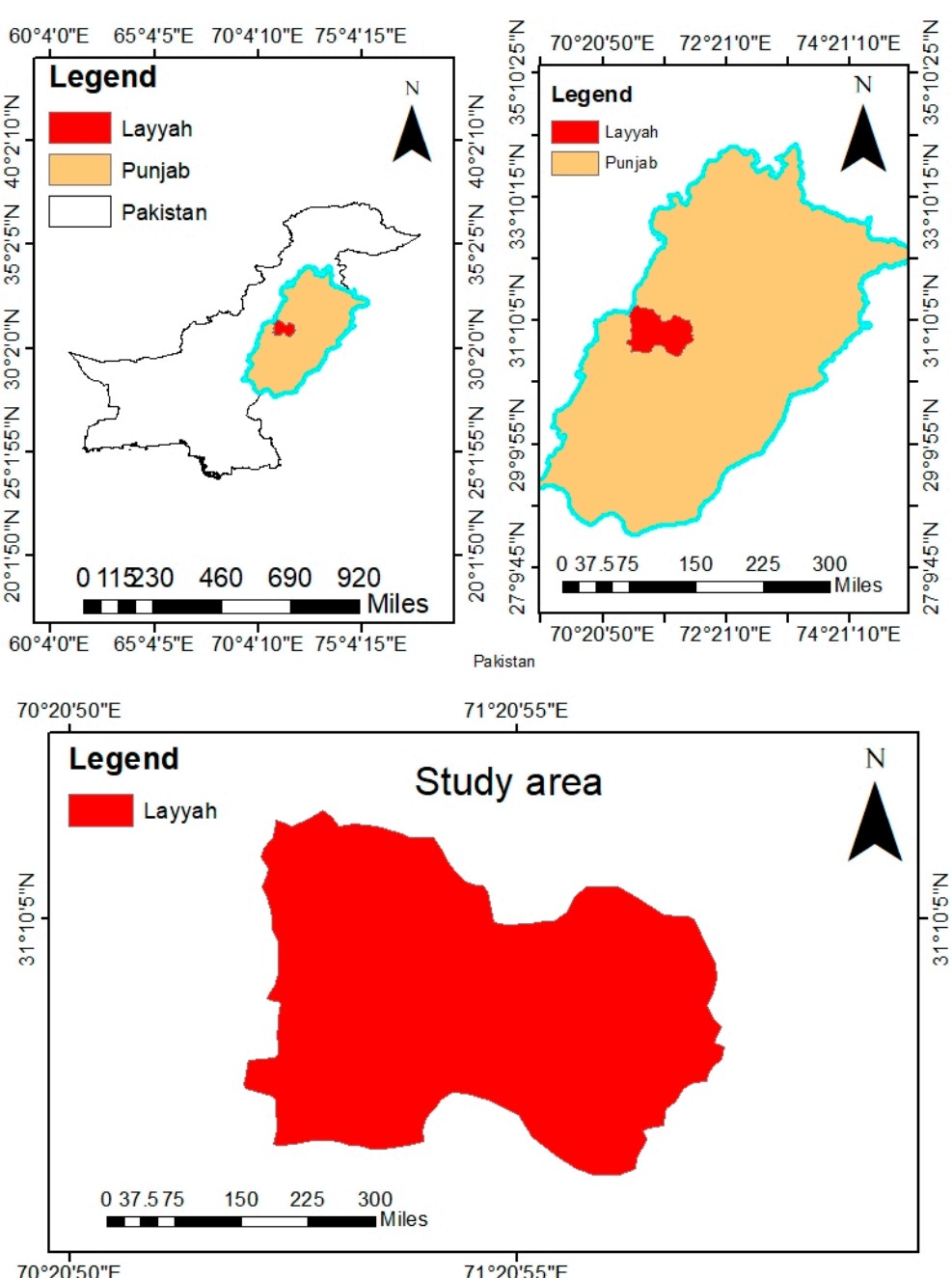

**Figure 1.** Map of Pakistan showing the wheat crop study region.

*2.2. Climate*

This trial was conducted in a subtropical area where medium rainfall occurs from July to September (in monsoon season) and low rainfall occurs during the rest of the season (in winter). Additionally, the months of March to June are hot and generally humid. The experimental site was situated at an altitude of 176 m above sea level. The Automatic Meteorological Station (AMS) installed at Layyah District, South Punjab, Pakistan, was approached to obtain the meteorological data, which included maximum and lowest temperatures as well as rainfall during the crop period (Figure 2). According to meteorological statistics, the mean maximum and lowest temperatures for both years 2019–2020 and 2020–2021 were 39.7 °C and 42.3 °C and 4.94 °C and 5.00 °C, respectively.

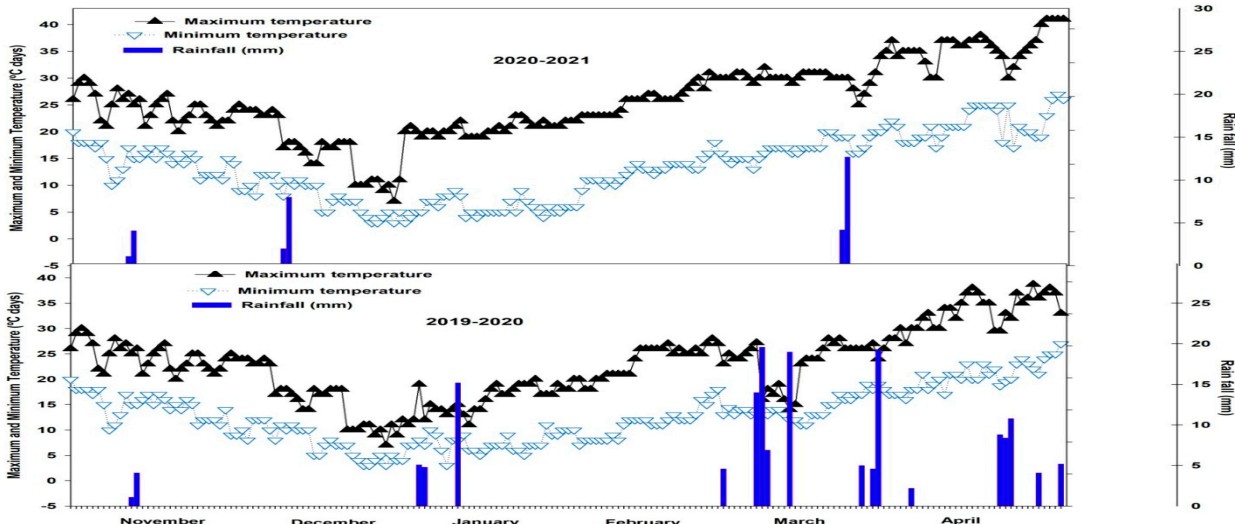

**Figure 2.** Weather graph showing maximum and minimum temperature and rainfall.

### 2.3. Design and Methods of Experiments

The present study was executed with three replicates in split-split plot arrangement. The net size of the experimental plot was kept 4 m × 1.8 m. Tillage treatments were arranged in the main plots, while sowing dates (15 November; 5 December; 25 December) were in sub-plots and boron levels (control, 2 kg ha$^{-1}$ soil applied, 2 kg ha$^{-1}$ foliar, water spray) were kept in sub-sub plots. The soil application of boron was performed 30 days after the sowing of wheat crop (tillering stage) and was applied near the rows of wheat crop. Similarly, the foliar application of boron and water spray was undertaken at the grain filling stage under both conventional and zero tillage systems.

### 2.4. Crop Husbandry

The experimental field was ploughed with the help of a tractor-mounted plough (moldboard plough) in conventional tillage and was leveled with the help of a laser leveler. The sowing of wheat crop under conventional tillage systems was carried out with the help of a manual hand drill. However, no cultivation practices were carried out in zero tillage and the sowing of wheat seeds was undertaken with the help of a seed-cum-fertilizer drill under zero tillage systems. The row-row spacing was kept at 22.5 cm, while the sowing depth of wheat seeds was kept 1.5 inches deep in both conventional and zero tillage systems. The wheat variety (Faisalabad-2008) was planted throughout two years of the field experimentation. P and K @ 60: 60 kg ha$^{-1}$ and N (60 kg ha$^{-1}$) from a total of 120 kg ha$^{-1}$ were applied following an optimum dose using fertilizer sources of diammonium phosphate (DAP), murate of potash and urea fertilizer, respectively, at planting time. The residual amount of N (60 kg ha$^{-1}$) was applied equally with third and fifth irrigations. Depending upon the environmental conditions and crop requirements, five irrigations were applied through proper irrigation scheduling depending on crop water requirements, especially at critical phenological stages of wheat crop. Weeds were controlled with herbicides application (Buctril Super 60% EC). However, an attack of fungus disease was seen on the late-sown wheat crop during both years and was controlled with spraying suitable fungicides.

### 2.5. Sampling and Measurements

Data on different crop traits (growth and yield) were recorded from tagged plants following the procedures given by Hussain et al. [10]. The plant height was recorded at full maturity of ten tagged plants using measuring tape and was computed to obtain the mean. The chlorophyll contents of ten tagged plants were recorded at crop maturity stage by using a chlorophyll meter (SPAD-502; Minolta, Tokyo, Japan). Moreover, ten spikes

were randomly selected at crop maturity from each experimental plot to determine spike length and spike weight. The number of grains was also recorded from ten selected spikes and their mean was determined. Similarly, the thousand grains' weight was determined by counting 1000 seeds from each experimental plot and weighing using an analytical weighing balance. At the crop harvest, the final biological, grain yield and harvest index were recorded by taking the samples of one square meter, and were then converted for the estimation of wheat crop yield per hectare. Grain boron contents were measured using Atomic Absorption Spectrophotometer. For this, 1 g of grain sample was taken, washed, dried and placed in microwave vessels. Afterwards, Perchloric acid (10 mL) and nitric acid (2 mL) were added to each grain sample. Then, the prepared solution was kept on a hot plate at 275 °C temperature for the digestion process until the color of the solution disappeared. The solution was filtered and its volume was diluted to 25 mL by adding distilled water. Then, the value of grain boron contents in the digested grains sample was recorded by using the Atomic Absorption Spectrometer

### 2.6. Statistical Analysis

The analysis of variance technique was used to determine the significance of the main effects and interactions for experimental treatments [40]. The means were differentiated by using the HSD test at 5% level of probability by using statistical software (Statistix-8.1).

### 3. Results

This study showed that different sowing dates and boron fertilization have a significant influence on plant height (PH), the number of tillers (NTPP), chlorophyll contents (CHCs), spike length (SL), spike weight (SW), 1000-grains weight (TGW), biological yield (BY), grain yield (GY) and the grain boron content (GBC) of wheat crops, at $p \leq 0.05$. However, the wheat crop was not affected by the different tillage practices. Furthermore, the results also showed that interactive influences of tillage systems with sowing dates and boron applications had no marked effects, and that the interactive effect of sowing dates with tillage systems also had no proper effects on wheat crop growth, yield and quality traits (Tables 1–3). Similarly, the interactions of tillage systems with sowing dates and tillage systems with boron were not significantly different during both years of field experimentation. A higher plant height, number of tillers and chlorophyll content were recorded in the 15 November sown crop compared to the late sown and control treatments. Similarly, the late-sown wheat crop with boron application also showed improved crop growth and yield parameters as compared to control treatments, where no boron application was made. Furthermore, boron foliar application (2 kg ha$^{-1}$) showed the highest morpho-physiological traits (Table 1). Maximum SL, SW, SPS, and NGPS were noted in the timely sown crop (15 November). However, the late-sown crop also showed improved spike length, spike weight, spikelets per spike, and number of grains per spike as compared to control treatments where no boron was applied. Additionally, the 1000-grains weight, biological and grain yield were also non-significant with the interactive effects of sowing dates with boron fertilization and different tillage systems (Table 3). Moreover, higher TGW, BY and GY, and BGCs were also recorded in the 15 November sown crop, but a significant improvement was also seen in the late-sown crop with the application of boron compared to the control treatment where no application of boron was carried out. The foliar-applied boron (2 kg ha$^{-1}$) also resulted in a higher crop productivity than the control, soil applied boron and water spray (Table 3).

**Table 1.** Response of *PH*, *NTPP* and *CHCs* to different sowing dates and boron applications under different tillage practices.

| Treatment | PH (cm) | | NTPP | | CHCs (SPAD) | |
|---|---|---|---|---|---|---|
| **Tillage Practices (TS)** | **2019–2020** | **2020–2021** | **2019–2020** | **2020–2021** | **2019–2020** | **2020–2021** |
| Zero Tillage | 80.5 | 82.6 | 330 | 332 | 43.2 | 45.3 |
| Conventional Tillage | 81.8 | 83.9 | 332 | 334 | 44.6 | 46.6 |
| LSD ($p \leq 0.05$) | 1.40 | 1.30 | 2.57 | 2.57 | 1.44 | 1.52 |
| Sowing Dates (SD) | | | | | | |
| 15 November | 82.6 A | 83.9 A | 357 A | 359 A | 46.0 A | 48.0 A |
| 5 December | 70.7 B | 83.8 A | 337 B | 339 B | 43.6 B | 45.6 B |
| 25 December | 64.7 C | 82.0 B | 311 C | 312 C | 42.2 C | 44.2 C |
| LSD ($p \leq 0.05$) | 3.44 | 1.23 | 3.89 | 3.90 | 0.98 | 0.96 |
| Boron levels (B) | | | | | | |
| $B_0$ | 79.2 D | 81.4 D | 332 D | 334 D | 41.4 D | 43.3 D |
| $B_1$ | 81.4 B | 83.6 B | 335 B | 337 B | 44.2 B | 46.4 B |
| $B_2$ | 83.8 A | 85.9 A | 338 A | 340 A | 47.5 A | 49.5 A |
| $B_3$ | 80.2 C | 82.1 C | 333 C | 335 C | 42.3 C | 44.7 C |
| LSD ($p \leq 0.05$) | 0.39 | 0.52 | 0.29 | 0.32 | 0.17 | 0.18 |
| TS | NS | NS | NS | NS | NS | NS |
| SD | ** | ** | ** | ** | ** | ** |
| TS × SD | NS | NS | NS | NS | NS | NS |
| B | ** | ** | ** | ** | ** | ** |
| TS × B | NS | NS | NS | NS | NS | NS |
| SD × B | ** | ** | ** | ** | ** | ** |
| TS × SD × B | NS | NS | NS | NS | NS | NS |

The mean values which are showing the same letters dp not differ at $p \leq 0.05$; ** = significant at $p \leq 0.01$; NS = not significant $p \leq 0.05$; $B_0$ = no application; $B_1$ = 2 kg ha$^{-1}$ (soil application); $B_2$ = 2 kg ha$^{-1}$ (foliar application); $B_3$ = water spray; PH = plant height; NTPP = number of tillers per plant; CHCs = chlorophyll contents.

**Table 2.** Response of *SL*, *SW*, *SPS* and *NGPS* to different sowing dates and boron under different tillage practices.

| Treatment | SL (cm) | | SW (g) | | SPS | | NGPS | |
|---|---|---|---|---|---|---|---|---|
| **Tillage Practices (TS)** | **2019–2020** | **2020–2021** | **2019–2020** | **2020–2021** | **2019–2020** | **2020–2021** | **2019–2020** | **2020–2021** |
| Zero Tillage | 10.9 | 11.2 | 1.79 | 1.81 | 10.7 | 11.7 | 20.5 | 22.7 |
| Conventional Tillage | 10.9 | 11.3 | 1.80 | 1.82 | 11.8 | 12.8 | 23.5 | 25.9 |
| LSD ($p \leq 0.05$) | 0.7 | 0.89 | 0.01 | 0.53 | 0.6 | 0.7 | 2.77 | 3.25 |
| Sowing Date (SD) | | | | | | | | |
| 15 November | 11.4 A | 11.6 A | 1.90 A | 1.92 A | 13.4 A | 14.3 A | 27.2 A | 29.1 A |
| 5 December | 10.9 B | 11.4 A | 1.76 B | 1.78 B | 11.2 B | 12.1 B | 22.3 B | 24.5 B |
| 25 December | 10.5 C | 10.8 B | 1.71 C | 1.74 C | 9.43 C | 10.4 C | 16.6 C | 19.3 C |
| LSD ($p \leq 0.05$) | 0.23 | 0.26 | 0.03 | 0.03 | 0.50 | 0.50 | 1.40 | 1.68 |
| Boron levels (B) | | | | | | | | |
| $B_0$ | 10.2 C | 10.5 C | 1.76 D | 1.79 D | 10.5 D | 11.5 D | 20.7 C | 23.6 B |
| $B_1$ | 11.1 B | 11.4 B | 1.79 B | 1.82 B | 11.5 B | 12.4 B | 22.4 B | 24.4 AB |
| $B_2$ | 12.1 A | 12.4 A | 1.82 B | 1.85 A | 12.3 A | 13.3 A | 23.2 A | 25.4 A |
| $B_3$ | 10.4 C | 10.8 C | 1.78 C | 1.81 C | 10.9 C | 11.8 C | 21.7 B | 23.8 B |
| LSD ($p \leq 0.05$) | 0.40 | 0.45 | 7.40 | 7.58 | 0.30 | 0.31 | 0.66 | 1.10 |
| TS | NS | NS | NS | NS | NS | NS | NS | NS |
| SD | ** | ** | ** | ** | ** | ** | ** | ** |
| TS × SD | NS | NS | NS | NS | NS | NS | NS | NS |
| B | ** | ** | ** | ** | ** | ** | ** | ** |
| TS × B | NS | NS | NS | NS | NS | NS | NS | NS |
| SD × B | ** | ** | ** | ** | ** | ** | ** | ** |
| TS × SD × B | NS | NS | NS | NS | ** | ** | NS | NS |

The values which have same letters are not significantly different at $p \leq 0.05$; ** = significant at $p \leq 0.01$; ns = not significant at $p \leq 0.05$; $B_0$ = no application; $B_1$ = 2 kg ha$^{-1}$ (soil application); $B_2$ = 2 kg ha$^{-1}$ (foliar application); $B_3$ = water spray; TS = tillage practices; SD = sowing dates; B = boron; SL = spike length; SW = spike weight; SPS = seeds per spike; NGPS = number of grains per spike.

**Table 3.** Effect of sowing date and boron application on *TGW*, *BY*, *GY* and *BGCs* under different tillage practices.

| Treatment | 1000-Grain Weight (g) | | Grain Yield (kg ha$^{-1}$) | | Biological Yield (kg ha$^{-1}$) | | Grains Boron Contents (mg kg$^{-1}$) | |
|---|---|---|---|---|---|---|---|---|
| Tillage Practices (TS) | 2019–2020 | 2020–2021 | 2019–2020 | 2020–2021 | 2019–2020 | 2020–2021 | 2019–2020 | 2020–2021 |
| Zero Tillage | 40.6 | 42.5 | 3949 | 4049 | 5170 | 5245 | 1.02 | 1.04 |
| Conventional Tillage | 41.3 | 43.2 | 3955 | 4052 | 5297 | 5394 | 1.03 | 1.05 |
| LSD ($p \leq 0.05$) | 0.71 | 0.90 | 16.7 | 27.9 | 52.2 | 55.8 | 0.99 | 0.01 |
| Sowing Date (SD) | | | | | | | | |
| 15 November | 41.3 A | 42.8 A | 4073 A | 4168 A | 5455 A | 5555 A | 1.03 A | 1.05 A |
| 5 December | 38.2 B | 42.2 A | 3918 B | 4018 B | 5322 B | 5418 B | 1.02 B | 1.04 B |
| 25 December | 34.6 C | 40.8 C | 3865 C | 3965 C | 4922 C | 4985 C | 1.02 B | 1.04 B |
| LSD ($p \leq 0.05$) | 1.98 | 0.98 | 38.2 | 38.6 | 112.2 | 134.8 | 0.01 | 0.01 |
| Boron levels (B) | | | | | | | | |
| B$_0$ | 39.8 D | 41.2 D | 3872 C | 3972 D | 5050 D | 5150 C | 1.01 C | 1.03 C |
| B$_1$ | 41.4 B | 42.8 B | 3939 B | 4034 C | 5175 C | 5275 B | 1.02 B | 1.04 B |
| B$_2$ | 43.5 A | 45.0 A | 4038 A | 4132 A | 5407 A | 5502 A | 1.04 A | 1.06 A |
| B$_3$ | 40.78 C | 42.1 C | 3959 B | 4064 B | 5302 B | 5352 B | 1.02 B | 1.04 B |
| LSD ($p \leq 0.05$) | 0.38 | 0.44 | 19.7 | 23.2 | 62.3 | 93.1 | 5.65 | 5.74 |
| TS | NS | NS | NS | NS | NS | NS | NS | NS |
| SD | ** | ** | ** | ** | ** | ** | ** | ** |
| TS × SD | NS | NS | NS | NS | NS | NS | NS | NS |
| B | ** | ** | ** | ** | ** | ** | ** | ** |
| TS × B | NS | NS | NS | NS | NS | NS | NS | NS |
| SD × B | ** | ** | ** | ** | ** | ** | ** | ** |
| TS × SD × B | NS | NS | NS | NS | NS | NS | NS | NS |

The values which have the same letters are not significantly different at $p \leq 0.05$; ** = significant at $p \leq 0.01$; NS = non-significant $p \leq 0.05$; B$_0$ = no application; B$_1$ = 2 kg ha$^{-1}$ (soil application); B$_2$ = 2 kg ha$^{-1}$ (foliar application); B$_3$ = water spray; TS = tillage practices; SD = sowing dates; B = boron.

## 4. Discussion

Micronutrients are important for the proper growth and development of crop plants, and their deficiency is a major growth limiting factor which leads towards a decrease in crop yield and production [16]. However, Pakistani soils are deficient in micronutrients, especially boron, which impairs grain setting in wheat resulting in declined number of grains per spike and ultimately reduced crop production [10,41]. Similarly, the optimum sowing date is also considered a major crop management factor, triggering the yield in cereal crops [42]. The sowing of wheat at proper sowing time is crucial under changing climatic conditions in a country like Pakistan, and delayed sowing will result in huge production losses [38]. Hence, the present study showed that different sowing dates and boron applications significantly affected the crop's morphological, physiological and yield parameters during years 2019–2020, and 2020–2021 (Tables 1–3).

In the current field study, the 15 November sown crop reflected considerably higher PH, NTPP and CHCs than the wheat crop sown on 5 December and 25 December, during both years of field experimentation. These results were highly correlated with the study of Fazily et al. [42], where they found that the sowing of wheat crop around 15 November, produced noticeably higher plant heights, effective tillers, days to physiological maturity, and grain yields than those of the late-sown wheat crop. This might be because of the optimum duration and suitable temperature for higher biomass accumulation, and ultimately more assimilate partitioning [8,42]. However, under late-sown conditions, the boron foliar application showed higher crop growth and yield parameters as compared to control treatments, which might be due to enhanced nutrient uptake and improved plant metabolic activities such as photosynthesis and transpiration [21].

Further, emerging evidence from other recent studies also showed that the late-sown wheat crop results in higher crop growth and yield with the application of mineral nutrients [8,43]. Our studied results also correspond to the study of Alam et al. [44], which documented that the late-sown wheat crop produced higher growth and yield traits due to an improved metabolism, nutrient uptake and protein synthesis as a result of boron application [18]. The maximum tillers, spikes and grain weight under late sowing ultimately

resulted in high grain and straw yields with boron application [45], which also supports our study. It might be associated with the results of [26], where improvements in the plant metabolism resulted in improved crop growth and yield components. Additionally, different studies' results have shown that the late planting of wheat crops adversely affects seed germination and other crop parameters, and can result in the reduced height of plants, and tillers/plant. However, boron application has made positive effects on these traits under late sown scenarios, due to which crop yield improved under late-sown conditions [10].

Boron improves crop quality and productivity, notably in the case of vegetables and fruits, through either direct or indirect methods. According to research findings, the foliar application of boron has the potential to improve the nutrient composition, productivity, and quality of a variety of agricultural produce. According to Nadeem et al. [46], boron application improved rice crop germination characteristics such as reduced time to germination, increased germination energy, higher ultimate germination percentage, lower mean germination time and improved germination index. Boron foliar spraying @ 2 kg ha$^{-1}$ has a significant impact on future growth phases. According to Nawaz et al. [14], the application of boron to the leaves had a significant beneficial effect on seed germination and seedling growth. Bielski et al. [26] conducted a study to examine the effects of boron treatment on several yield-related parameters, such as seedling emergence, leaf appearance and elongation, chlorophyll content, water relations and yield. According to Islam et al. [18], the use of boron resulted in a significant increase of 64% in grain yield compared to the control group. However, when considering the amount of boron present in the grain, the increase in crop yield was only 27%.

Similarly, higher pH and CHCs were noted with the foliar application of boron @ 2 kg ha$^{-1}$. According to the research conducted by Tahir et al. [11], wheat that received foliar sprays of boron at any developmental stage such as tillering, jointing, booting, and anthesis produced more grains per spike when compared to no boron fertilization, because B is involved in at least sixteen essential plant processes. Therefore, improved yield traits were obtained when the wheat crop was late planted with the foliar application of boron [11,47]. In this context, our studied results also corresponded to a study carried out by [48], which showed that foliar-applied boron at 2 kg ha$^{-1}$ improved the grain yield of the late-sown wheat crop. It was also explained by [26] that boron application (1.6 kg ha$^{-1}$) improved the wheat grain yield and yield components. The positive improvements in yield components resulted in improved crop production [26]. Similarly, different studies by [47,49] also support our study findings because they found that the foliar application of boron improves late-sown wheat crop growth and productivity. Improvements in crop growth and productivity might be associated with improved plant metabolism by the application of boron. This might be due to improved metabolism, nutrient uptake and protein synthesis with boron application [18]. Further, boron application also reduces the impacts of heat stress on late sown wheat crop, which results in enhanced crop productivity under late-sown conditions [30]. However, the wheat crop was no longer affected by different tillage practices. Furthermore, the zero tillage system was more effective than a conventional tillage system in terms of carbon sequestration, greenhouse gas emissions, and N$_2$O emissions [14,35]. However, it takes a longer period of time (4–5 years) to enhance the crop growth and yield parameters by improving soil health. Due to this reason, in the present study the zero tillage system did not considerably affect crop productivity during either year.

## 5. Conclusions

The present investigation demonstrated that wheat crops that were sown on November 15th and received a boron foliar application (@ 2 kg ha$^{-1}$) gave higher crop growth, yield components, and boron content in the grains. The boron application @ 2 kg ha$^{-1}$ improved the growth and productivity of wheat crops that were sown late (on 5 December and 25 December). This improvement was observed in a comparison with the control treatments, where no boron was applied during delayed sowing. Therefore, the foliar ap-

plication of boron is a useful option to achieve maximum and sustained crop productivity under late-sown conditions. During the present investigation, however, different tillage systems had no significant effect on wheat crop growth and yield parameters.

**Author Contributions:** Conceptualization, M.I., S.U.-A. and A.S. (Ahmad Sher); methodology, M.I., S.U.-A. and A.S. (Ahmad Sher); software, K.M.; formal analysis, A.S. (Abdul Sattar), M.I. and S.A.; investigation, M.I. and F.R.; resources, A.S. (Abdul Sattar), S.U.-A. and M.I.; data curation, Y.A. and F.R.; writing—original draft preparation, M.I., S.M.S., I.H. and S.U.-A.; writing—review and editing, B.I., I.H. and Y.A.; supervision, S.U.-A. and A.S. (Ahmad Sher); project administration, M.I. and Y.A.; funding acquisition, S.A. All authors have read and agreed to the published version of the manuscript.

**Funding:** This research was funded by the Researchers Supporting Project number (RSP2023R194), King Saud University, Riyadh, Saudi Arabia.

**Institutional Review Board Statement:** Not applicable.

**Informed Consent Statement:** Not applicable.

**Data Availability Statement:** Not applicable.

**Acknowledgments:** We gratefully acknowledge the support from all authors and researchers. The authors would like to extend their sincere appreciation to the Researchers Supporting Project number (RSP2023R194), King Saud University, Riyadh, Saudi Arabia.

**Conflicts of Interest:** The authors declare no conflict of interest.

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
