# Peer review of "Boron Fertilization Alleviates the Adverse Effects of Late Sowing in Wheat under Different Tillage Systems"

_agriculture, doi:10.3390/agriculture13061229_

Round 1

Reviewer 1 Report

As a whole, an appreciable effort has been made by the authors to explore the crucial role of boron in alleviation of adversities related to late sowing in wheat under different tillage systems. However, some improvements should be made to enhance the overall quality of the manuscript in order to make it more attractive and understandable.

Overall, the English language requires a moderate improvement to establish the work as more scientific.

Author Response

We are thankful to worthy reviewer for the comments and especially good suggestions for the improvement of manuscript. Now, the extensive revision of the manuscript has been done while suggestion and comments have been incorporated in Track changes as guided by the worthy reviewer.

  1. Sentence structure, grammar, and short sentences are made as guided by the reviewer.
  2. We have also improved the writing style and language of the article. Article has also been edited for English language and grammar as guided.

Reviewer 2 Report

Comments:

The authors relied on 2-year research results, while the standard of field research is 3-year experiments. In this case, a certain justification is the similar research results in both years.

The experience was assumed to be 3-factor. The obtained results only confirmed the importance of the date of plant sowing, in this case winter wheat, which is commonly known in every region (country). The use of boron also turned out to be important, although the yield increase, despite statistical differences, is small.

The authors did not note any interactions between research factors. The reason for this is the incorrect assumption of the hypothesis - whether an interaction between plowing or no plowing and the timing of wheat sowing and the use of boron can be expected.

Questions to clarify or complete at work:

- boron was used in the tillering phase - can it be called soil fertilization?

- whether there was a shortage of boron in the soil - lack of soil analyzes for the content of this component and other microelements

The methodology needs to be supplemented:

- specify the name of the active substance of Buctril

- supplement agricultural technology with the names of preparations and their active substances used to combat diseases

- describe in more detail wheat sowing in the no-plough system, what were the parameters of the seeder, type of coulter, width of sowing, width of the cultivated strip of soil, etc.

- the yield was harvested from 1 m2 - the results should not be converted to 1 ha, it is too small a harvest area and the result should be given as the yield weight from 1 m2

- tables are difficult to read, LSD values are not included if the difference is insignificant, only "NS - as no significant differences)" - this is a common practice in scientific journals

- it is unnecessary to include both LSD values (if they are significant) and letter homogeneous groups at the same time - one of these variants should be selected

Author Response

(The authors gave the same response as above.)

Reviewer 3 Report

The publication entitled "Optimum Boron Fertilization Alleviates the Adverse Effects of Late Sowing in Wheat under Different Tillage Systems " presents important research results for science and agricultural practice. However, the manuscript needs improvement. I included detailed comments in the original text (pdf). See Annex. After making corrections, the article may be published in the journal Agriculture.

My most important comments:

save the units as required by the journal
add to keywords: foliar fertilization
improve the aim of research
correct and complete the missing information in the Material and Methods section,
- e.g. how did you analyze the boron in the grain
- at what development stage did you measure the SPAD, etc

check the statistical calculations and write what program you used
describe tables 1, 2 and 3 in more detail. The Results section is important
explain any abbreviations used

I hope that my comments will allow the authors to improve the text of the manuscript

Author Response

(The authors gave the same response as above.)

Reviewer 4 Report

The paper is really interesting and the hypothesis and the results are giving now information upon boron micronutrient deficiency. However there are some points where the pepar is to be improved.

1. There is less information in the literature review of the paper about the boron treatments and earlier findings in this field. The deficiency symtomps are discussed, however no previous examples are presented.

2. Further explanations would be needed to highlight the doses and the applications of boron on wheat crop. Why these doses? Why not bigger ones? Also, the application should be detailed about the means and the way of giving that to the soil and spraying on the crop.

3. In the materials and methods more detailed description of the trial would be needed regarding the plot size, the experimental layout, soil properties etc.

4. The agronomic part is to be cleared as well. Such terms like "tractor mounted plough" would not give enough information. What plough? A mouldboard one, a chisel one etc? What was the technigóque of no tillage. What do they mean on hand drill? row with, space, sowing depth etc are needed.

5. More accurate composition of applications is needed. For example SPAD values are giving information whenever repeated consequetly about chlorophyll conten in comparison with the earlier stages, but do not represent a stable measurement value.

6. The results are promising, however the interactions are to be evaluated as well.

I beieve that after these corrections and changes the paper would provide valuable information to the readers.

The paper is written in good English, but a thorouh check and a grammatical review would make a benefit.

Author Response

(The authors gave the same response as above.)

Round 2

Reviewer 2 Report

I believe the manuscript has been sufficiently improved.

Only in Tables 1-3 I still consider it unnecessary to provide the LSD value for "Tillage practices" since it is an insignificant difference (NS). Just write "NS" on this line instead of typing "NS" again on another line - it's confusing. Such a record is not found in other publications. I leave the decision to leave the Authors' version to the Editors.

Author Response

General comments

We are thankful to worthy reviewer for the comments and especially good suggestions for the improvement of manuscript. Now, the extensive revision of the manuscript has been done while suggestion and comments have been incorporated in Track changes as guided by the worthy reviewer.

Specific comments

Sr. No#

Comment/suggestion

Response

1

I believe the manuscript has been sufficiently improved.

Thank you very much for your appreciation.

2

Only in Tables 1-3 I still consider it unnecessary to provide the LSD value for "Tillage practices" since it is an insignificant difference (NS). Just write "NS" on this line instead of typing "NS" again on another line - it's confusing. Such a record is not found in other publications. I leave the decision to leave the Authors' version to the Editors.

Thank you for your valuable suggestion. All interactions have been corrected with the word "NS". Additionally, the reviewers have suggested that the word "NS" be incorporated in all three tables using the same format, which has been implemented accordingly. To clearly demonstrate the difference between row and column interaction, it is better to write "in each row."  

Reviewer 4 Report

The paper has been improved by the authors in accordance with the remarks. There are only two points where a more detailed explanation would be needed.

1. The interctions - however the authors reported that they were not significant - are to be presented.

2. The conclusions should be written in a more simple/humble way.

It seems to be appropriate by now, however prior to submission a final control would make a benefit.

Author Response

General comments

We are thankful to worthy reviewer for the comments and especially good suggestions for the improvement of manuscript. Now, the extensive revision of the manuscript has been done while suggestion and comments have been incorporated in Track changes as guided by the worthy reviewer.

Specific comments

Sr. No#

Comment/suggestion

Response

1

The paper has been improved by the authors in accordance with the remarks.

Thank you very much for your appreciation.

2

The interactions - however the authors reported that they were not significant - are to be presented.

Thank you for your valuable suggestions. All of the interactions have been corrected. The "NS" relationship has been included in all interactions to clearly distinguish between the various study parameters. 

3

The conclusions should be written in a more simple/humble way.

Thank you for your kind suggestion. The conclusion has been written in a more simple/humble way.